# Caldera resurgence during the 2018 eruption of Sierra Negra volcano, Galápagos Islands

Andrew F. Bell [1✉], Peter C. La Femina [2], Mario Ruiz [3], Falk Amelung[4], Marco Bagnardi [5], Christopher J. Bean[6], Benjamin Bernard [3], Cynthia Ebinger [7], Matthew Gleeson [8], James Grannell[6], Stephen Hernandez [3], Machel Higgins[2], Céline Liorzou[9], Paul Lundgren[10], Nathan J. Meier[2], Martin Möllhoff [6], Sarah-Jaye Oliva[7,12], Andres Gorki Ruiz[2] & Michael J. Stock [11]

Recent large basaltic eruptions began after only minor surface uplift and seismicity, and resulted in caldera subsidence. In contrast, some eruptions at Galápagos Island volcanoes are preceded by prolonged, large amplitude uplift and elevated seismicity. These systems also display long-term intra-caldera uplift, or resurgence. However, a scarcity of observations has obscured the mechanisms underpinning such behaviour. Here we combine a unique multi-parametric dataset to show how the 2018 eruption of Sierra Negra contributed to caldera resurgence. Magma supply to a shallow reservoir drove 6.5 m of pre-eruptive uplift and seismicity over thirteen years, including an Mw5.4 earthquake that triggered the eruption. Although co-eruptive magma withdrawal resulted in 8.5 m of subsidence, net uplift of the inner-caldera on a trapdoor fault resulted in 1.5 m of permanent resurgence. These observations reveal the importance of intra-caldera faulting in affecting resurgence, and the mechanisms of eruption in the absence of well-developed rift systems.

[1] School of GeoSciences, University of Edinburgh, Edinburgh, UK. [2] Department of Geosciences, The Pennsylvania State University, State College, PA, USA. [3] Instituto Geofísico, Escuela Politécnica Nacional, Quito, Ecuador. [4] Department of Marine Geosciences, University of Miami, Coral Gables, FL, USA. [5] Cryospheric Sciences Laboratory, NASA Goddard Space Flight Center, Greenbelt, MD, USA. [6] School of Cosmic Physics, Dublin Institute for Advanced Studies, Dublin, Ireland. [7] Department of Earth and Environmental Sciences, Tulane University, New Orleans, LA, USA. [8] Department of Earth Sciences, University of Cambridge, Cambridge, UK. [9] Laboratoire Géosciences Océan, Université de Bretagne Occidentale, Brest, France. [10] Jet Propulsion Laboratory, California Institute of Technology, Pasadena, CA, USA. [11] Department of Geology, Trinity College Dublin, Dublin, Ireland. [12] Present address: Department of Earth, Ocean and Atmospheric Sciences, University of British Columbia, Vancouver, BC, Canada. ✉email: a.bell@ed.ac.uk

asaltic calderas are the sites of some of the largest defor-
mation events on Earth. Eruptions at Kilauea in 2018
(refs. [1,2]), and Bárðarbunga in 2014 (refs. [3,4]), involved
rapid subsidence along caldera-bounding ring faults of >500
(ref. [5]) and 65 m (ref. [4]), respectively. However, the minor pre-
eruptive uplift and seismicity preceding these eruptions provided
little warning of the scale of these forthcoming events. In contrast,
some basaltic calderas in the Galápagos Islands, Ecuador, exhibit
several metres of uplift and intense seismicity for years before
eruptions[6–8]. In addition, elevated intra-caldera topography indi-
cates that these are sites of long-term uplift or resurgence[7,9,10],
rather than subsidence. Although resurgence is well documented
over long time scales at silicic calderas[11], it is rare at basaltic sys-
tems, and suggests fundamental differences between Galápagos
calderas and archetypes in Hawaii and Iceland.

The steep upper flanks, large calderas[8,12,13], and circumfer-
ential and radial eruptive fissures[14,15] of the volcanoes of the
western Galápagos Islands distinguish them from more inten-
sively studied basaltic systems in Hawaii and Iceland. This dis-
tinction has been furthered by observations from satellite data of
prolonged surface uplift and trapdoor faulting[6,16]. Galápagos
Island volcanoes have generally not developed subaerial rift
zones, like systems in Hawaii or the Canary Islands, which exert a
first-order control on shallow magma migration and emplace-
ment, and subsequent caldera formation. In addition, the wide
calderas and shallow depths of magma reservoirs of Sierra Negra
and Alcedo result in low roof thickness to diameter ratios[10,17,18].
However, despite their significance, the Islands' remote location
means that no previous eruption has been monitored by a local,
ground-based, geodetic, and seismic network. Consequently,

there are no multidisciplinary studies of the volcanic processes
underpinning Galápagos Island volcanism. Integration of geo-
detic, seismic, and geochemical observations through the eruption
cycle of a Galápagos caldera offers the possibility of new insights
into critical processes, thus starting to fill a major gap in our
understanding of basaltic volcanism.

Sierra Negra is an 1100 m high shield volcano on Isabela Island,
Galápagos (Fig. 1A). It has a large elliptical summit caldera, 9.5 km
by 7.5 km wide and 100 m deep[19,20] (Fig. 1B). Past eruptions
occurred on both circumferential and radial fissure systems, with
evidence for topographic control of dike orientation[14,21]. Geodetic
observations indicate the presence of a flat-topped sill-like magma
reservoir at ~2 km depth below the caldera[6,7,22], giving a low roof
thickness to caldera diameter aspect ratio of ~0.25 (compared to
>1.0 for Kilauea and Bárðarbunga). The sill's lateral extent coin-
cides with a 14 km long, C-shaped intra-caldera fault system,
known as the trapdoor fault (TDF)[6,23–25]. The TDF has produced
a 150 m high sinuous ridge[24] that rises above the south-western
caldera rim (Fig. 1C), indicating resurgence since the caldera
forming event. The mapped extent of the TDF system suggests the
trapdoor is hinged in the northeast (NE). Large earthquakes are
known to originate at Sierra Negra during the onset of recent
eruptions[26–28]. The southern segment of the TDF ruptured twice
prior to the previous eruption in October 2005 (ref. [29]), generating
earthquakes of Mw4.6 five months[6] and Mw5.5 three hours[7]
before the eruption onset.

On 26 June 2018, a new eruption began on the northern rim of
the caldera. Within 24 h, the fissure system extended 8.5 km to
the northwest (NW; Fig. 1B). By the end of the eruption on
23 August 2018, lava flows covered an area of 30.6 km² (ref. [28]).

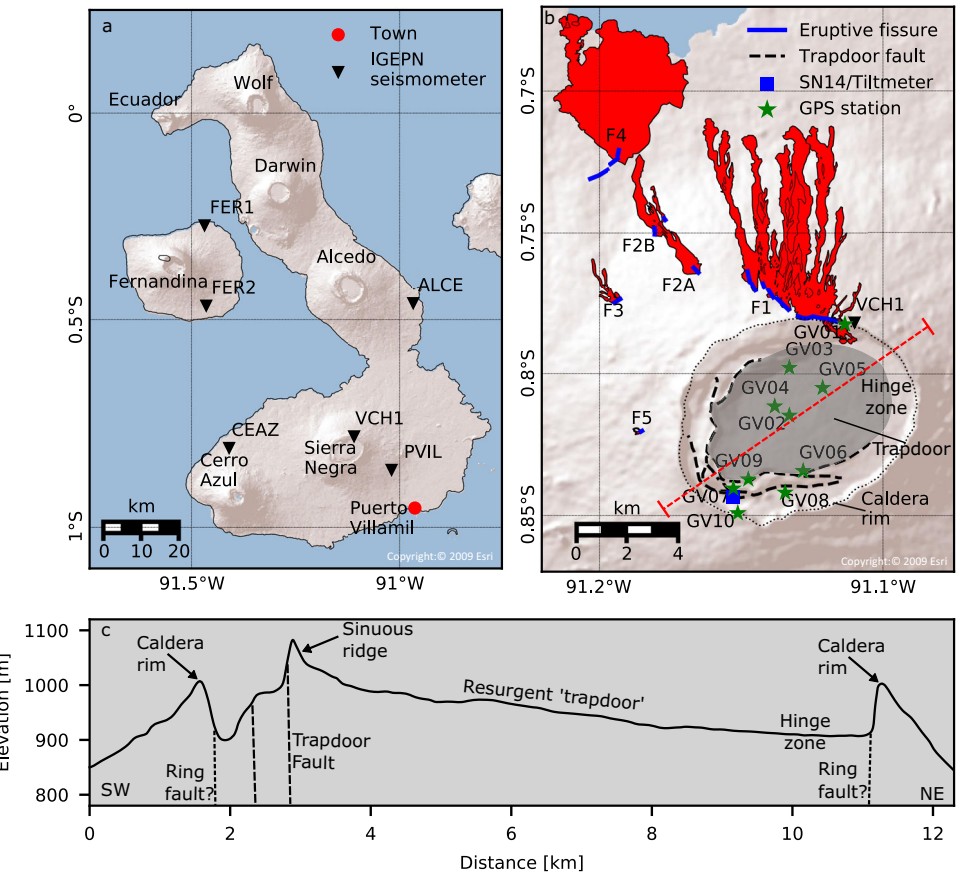

**Fig. 1 Location of Sierra Negra volcano. A** Volcanoes of Isabela and Fernandina Islands. **B** Vicinity of Sierra Negra volcano, showing geodetic network,
trapdoor fault system, lava flows (red), and eruptive fissures of the 2018 eruption. **C** SW–NE topographic profile across the caldera, along line indicated in **B**.

A ground-based monitoring network recorded the eruption, consisting of the permanent 6-station seismic network of the Instituto Geofísico, Escuela Politécnica Nacional (IGEPN; Fig. 1A), the 14-station IGUANA seismic network deployed in April 2018 (Supplementary Fig. 1), and a permanent 10-station continuous Global Positioning System (cGPS) network (Fig. 1B). Interferometric synthetic aperture radar (InSAR) remotely observed surface deformation. Samples of lava and tephra were collected during and after the eruption, and analysed for their geochemistry. These datasets constitute the first local multi-parametric observations of an eruption in the Galápagos Islands. They allowed the IGEPN to track the evolution of unrest and communicate it to local authorities and public.

Here, we show that the highest rate of pre-eruptive uplift and seismicity observed at a basaltic caldera occurred before the 2018 eruption of Sierra Negra. The caldera was uplifted >6.5 m in the 13 years preceding the eruption, driven by magma accumulation at 2 km depth, and stressing the intra-caldera TDF. The eruption was initiated by co-seismic uplift on the TDF, resulting in the failure of the sill and extra-caldera magma migration. Co-eruptive subsidence exceeded pre-eruptive uplift by ~2 m (i.e., 8.5 m). However, the caldera floor was uplifted a net 1.5 m indicating co-eruptive resurgence, distinguishing this eruption from recent basaltic caldera forming eruptions.

## Results

**Caldera inflation and intra-caldera seismicity.** The 2018 eruption involved what we understand to be the greatest surface uplift and seismicity ever observed during the pre-eruptive inflation phase at a basaltic volcano. Uplift initiated immediately after the end of the 2005 eruption[7,29], but the rate was not constant during the 13-year lead up to the 2018 eruption (Fig. 2A), with at least four distinct uplift episodes (phases 1–4 in Fig. 2) and one subsidence episode ('Methods'). From late 2017 until the onset of the eruption, during the latter stages of the fourth uplift phase, uplift rates remained constant, with the centre of the caldera uplifting at 1.4 m/year. The spatial distribution of uplift (Fig. 2C) is consistent with elastic deformation driven by an inflating flat-topped sill located at 2.0 km depth[30], similar to that observed for pre-eruptive inflation before the 2005 eruption[7,29]. In total, >6.5 m of uplift was observed in the centre of the caldera (Fig. 2A), equating to a total inferred increase in sill volume of 0.21 km³ ('Methods').

The earliest seismic data recorded at Sierra Negra[31] indicate that seismicity was already elevated (relative to rates after the 2018 eruption) by late 2009. Rates of earthquakes ('Methods') increased during an uplift pulse in 2013–2014, and then rapidly from late 2016 (Fig. 2B). From late 2017 until the start of the eruption, the high rate remained constant (Fig. 3B). Between 1 January 2018 and 26 June 2018, there were 12 > M4.0 earthquakes, including an mb4.4 (USGS) event on 14 March. There is no evidence that these events caused a change (increase or decrease) in uplift rate. Located earthquakes ('Methods') recorded from April 2018 are almost entirely restricted to the TDF at depths of 3 km or less. They define a nearly complete ring structure, but with gaps in seismicity in the NE (the hinge) and NW (Fig. 3C). There is no systematic evolution of the location of these pre-eruptive earthquakes along the TDF with time (Supplementary Video 1). Focal mechanisms ('Methods') indicate reverse faulting in the west and NW, normal faulting in the NE

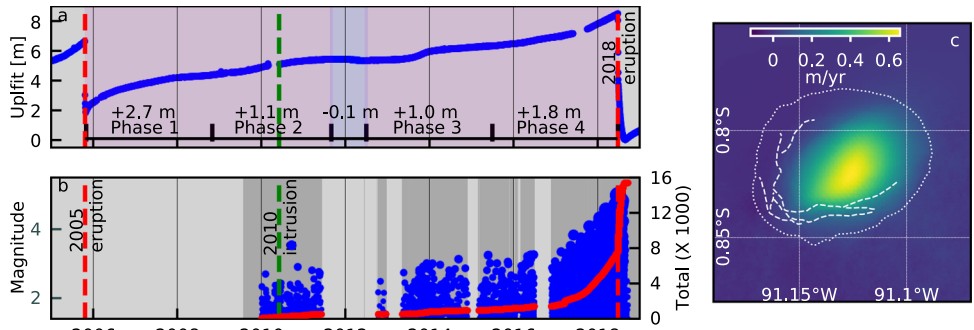

**Fig. 2 Deformation and seismicity during the pre-eruptive phase at Sierra Negra volcano. A** Vertical component of deformation recorded at cGPS stations GV02 and GV04 ('Methods'). Vertical dashed lines indicate timing of the 2005 eruption (red), 2010 deep intrusion (green), and 2018 eruption (red). **B** Estimated earthquake magnitudes (blue circles) and total number of *M* > 1.4 earthquakes recorded at VCH1 (red line). **C** Radar line-of-sight average velocity between 13 March 2014 and 07 June 2018 from Sentinel-1 InSAR ('Methods').

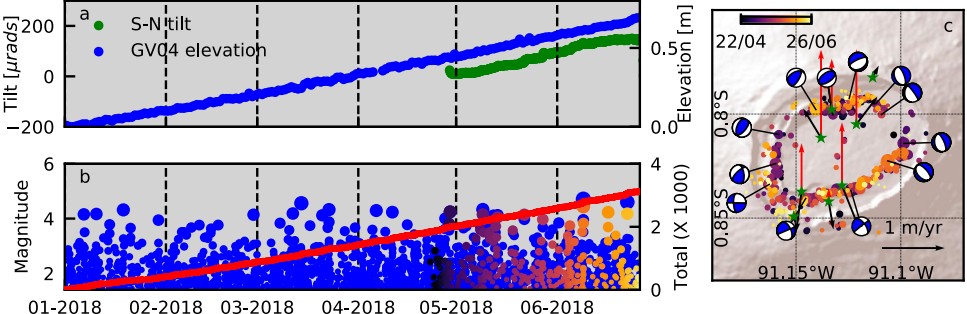

**Fig. 3 Deformation and seismicity during final pre-eruptive phase (01 January 2018–26 June 2018) at Sierra Negra volcano. A** Uplift recorded at GV04 and tilt (positive to the S) recorded at SN14 (Fig. 1B); **B** Magnitudes of earthquakes (circles) and total number with *M* > 1.4 (red line) detected at VCH1 (blue) and located by the IGUANA network (coloured by time). **C** Epicentres of earthquakes, lower-hemispherical projections of focal mechanisms, and average horizontal (black) and vertical (red) displacement rate vectors for 22 April 2018–26 June 2018.

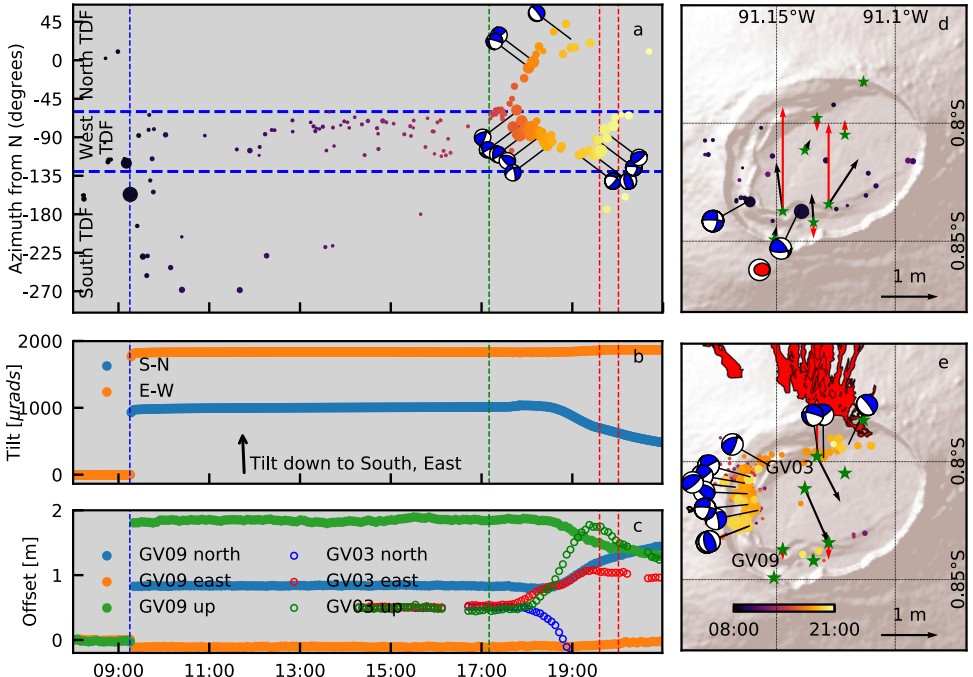

**Fig. 4 Seismicity and deformation associated with the onset of eruption on 26 June 2018. A** Azimuth of earthquake epicentres from centre of caldera and focal mechanisms of earthquakes as a function of time (UTC). Circle size indicates magnitude, as in Fig. 3B. Vertical dashed lines indicate timing of the Mw5.4 earthquake (blue), onset of seismic swarm (green), and onset of tremor pulses (red). **B** Tilt recorded at station SN14 (for tiltmeter location, see Fig. 1E). **C** Deformation recorded at cGPS stations GV09 and GV03. Before 09:15, all components plot at zero. **D** Earthquake epicentres, focal mechanisms, and horizontal (black) and vertical (red) cGPS displacements between 08:00 and 12:00, including the Mw5.4 earthquake. Red focal mechanism is the teleseismic Harvard CMT solution. **E** as **D** for the period 12:00–21:00. Extent of lava flows shown in red.

and east, and strike-slip faulting in the south (Fig. 3C). The spatial distribution of hypocentres indicates that seismicity occurred on steep fault planes, but do not constrain the dip direction.

**Pre-eruptive trapdoor faulting and eruption onset.** The onset of the 2018 eruption involved a previously unsuspected complex interplay between intra-caldera faulting and multi-directional shallow magma migration. At 09:15 26 June 2018 (all times UTC), an Mw5.4 earthquake ruptured the southern section of the TDF (Fig. 4A). The earthquake resulted in 1.83 m uplift at cGPS station GV09 located on the hanging wall of the TDF (Fig. 4C, D), and 0.26 m subsidence at GV08 south of the TDF. SN14, located on the southern footwall of the TDF, tilted by >2000 micro-radians down to the southeast (Fig. 4B). Neither cGPS nor tilt data showed any precursory signal in the hours before the earthquake. Surface ruptures were observed along the southern TDF. The co-seismic uplift of the trapdoor is consistent with the USGS and Harvard CMT teleseismic moment tensors (a steeply northward (inward) dipping reverse fault). The IGUANA-derived focal mechanism also suggests reverse faulting, though with an oblique slip component (Fig. 4D). Modelling results suggest that the Mw5.4 earthquake caused a normal stress change of >0.1 MPa (1 bar) at a depth of 2 km on the margins of the caldera system (Supplementary Fig. 3). However, this stress change did not immediately initiate reservoir failure. Low rates of seismicity and uplift persisted for 8 h. A further 50 microradians of tilt and 5–10 cm of uplift suggest a pulse of post-seismic inflation of the sill or slip on the TDF (Supplementary Fig. 2).

At 17:00, a seismic swarm initiated near the NW corner of the TDF, and earthquake rates and magnitudes increased rapidly. Between 17:45 and 19:00 epicentres migrated in two directions: eastwards along the northern TDF, towards the location of the initial eruptive fissure; and southwards along the western TDF (Fig. 4A, E and Supplementary Video 2). Focal mechanisms are

similar to those during the preceding 2 months, suggesting that they result from near-vertical displacement of the TDF. GV03 and GV04 recorded south-eastward movement and uplift from 17:35 to 19:25 (Fig. 4C, E), consistent with a dike ascending in or near the northern TDF and progressively propagating eastwards. The southern trapdoor (GV06) started to subside slowly at 17:35. cGPS and seismic observations do not constrain the geometry of the initial intrusion in the west of the caldera, but stations in the southwest corner of the trapdoor responded first as seismicity migrated southwards at 17:40, before moving rapidly back to the north at 18:30 as subsidence accelerated (Fig. 4B, C). The eruption began at 19:35, as indicated by seismic tremor, at fissure F1 just outside the northern caldera rim[28]. An infrasound signal, recorded at station IS20 on nearby Santa Cruz Island, also marked the eruption onset. A second, higher amplitude, tremor pulse began at 20:00.

**Co-eruptive deformation and seismicity.** Almost 8.5 m of subsidence and intense seismicity accompanied the eruption, but most of the permanent uplift on the TDF was not recovered. The eruptive fissure system opened aseismically to the NW, with fissures F1–3 (Fig. 1B), and probably F5, active on the night of 26 June. The first 24 h of the eruption accounted for ~50% of the erupted volume[28]. Initial co-eruptive deflation was correspondingly rapid, with ~300 cm (~12.5 cm/h) vertical subsidence at GV04 in the centre of the caldera in the first 24 h (Fig. 5A). High rates of seismicity (up to Ml4.3) accompanied this subsidence, with locations distributed along the TDF (Fig. 5C).

On 27 and 28 June, a distinct swarm of earthquakes with Ml < 2.3 was located in the west flank (Fig. 5C). Although the location uncertainties for these events are large, they coincide with surface deformation observed in InSAR images showing a second distinct shallow intrusion to the NW (Fig. 5D). Analysis of ALOS-2 data indicate ~2 m of uplift associated with the intrusion during the

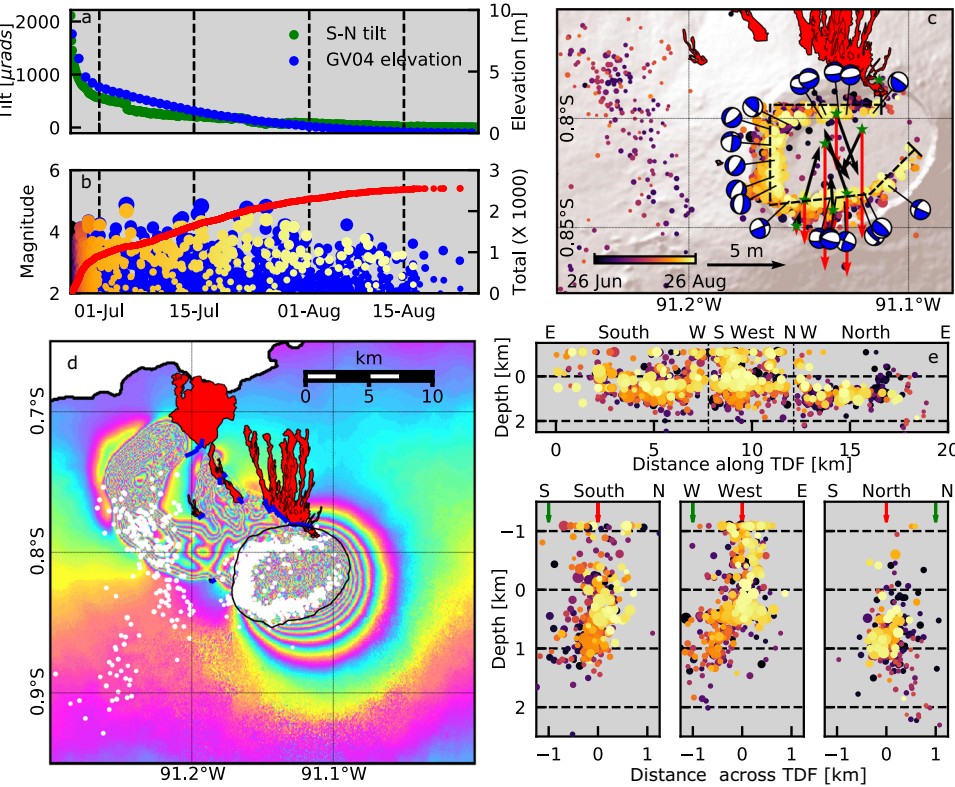

**Fig. 5 Deformation and seismicity data associated with the eruption of Sierra Negra between 26 June 2018 and 26 August 2018. A** Vertical displacement at GV04 and tilt recorded at SN14. **B** Magnitudes of earthquakes detected at VCH1 (blue), and located by the IGUANA network (coloured by time, size corresponding to magnitude), and total number of recorded earthquakes $M > 2.0$ (red). **C** Earthquake epicentres, focal mechanisms, and cGPS displacements during this interval. **D** ALOS-2 interferogram between 18 May 2018 and 29 June 2018 ('Methods'). Each colour cycle represents ~11.5 cm of displacement in the line-of-sight direction, consistent with subsidence over the caldera and uplift over intrusions in the NW flank. Lava flows in red, eruptive fissures in blue. Locations of co-eruptive earthquakes shown with white circles, caldera rim indicated with black line for reference. **E** Earthquake hypocentral depth as a function of distance along the TDF (top panel) and from the surface expression of the southern, western, and northern limbs of the TDF (red arrows). Surface location of the caldera rim (green arrows) for reference. Colours and sizes of circles as **B** and **C**. Line of section shown in **C**.

propagation phase. This intrusion further propagated to the north, feeding a second eruptive phase that began at fissure F4 on 1 July, lasting until 25 August. Subsidence continued but at a lower rate.

Seismicity continued throughout the eruption, including 14 > Ml4.0 events. Rates progressively decreased with time, correlated with the decrease in subsidence rate. Co-eruptive epicentres depict an almost complete ring, with the only gap associated with the hinge in the NE. Focal mechanisms of large earthquakes (Fig. 5C) indicate a reactivation of the TDF, but with a reversal of slip directions from the inflationary phase. All seismicity was located above the sill (Fig. 5E). An Mw5.0 earthquake on 5 July on the southern TDF had opposite displacement to the 26 June Mw5.4 event, displacing GV06 and GV09 by 71 and 15 cm, respectively. Large earthquakes did not change the subsidence rate.

The eruption ended on 25 August 2018 (ref. [28]). In total, there was 8.48 m of co-eruptive subsidence in the central caldera (Fig. 2A), exceeding the pre-eruptive uplift of 6.5 m, and over 2000 microradians tilt down to the north. The estimated erupted volume of $0.141 \pm 0.071$ km$^3$ DRE[28] matches well with the estimate of magma accumulated during pre-eruptive inflation (0.21 km$^3$). This correlation is not always observed in the Galápagos; during the 2015 eruption of Wolf volcano, the deflation of a shallow source accounted for only ~5% of the emitted magma, with the majority being supplied directly from a deep source[32].

As in 2005, caldera uplift re-initiated immediately after the end of the 2018 eruption. At the time of writing uplift has exceeded ~1.5 m (Fig. 2A), and low magnitude earthquakes are occurring at rates ~1/4 of those observed in late 2009.

**Petrology of erupted products**. Lava compositions vary systematically through the 2018 eruption and indicate that it involved magma from two different sources. All erupted magmas are near-aphyric basalts, similar to previous Sierra Negra eruptions (Supplementary Fig. 4), but extending to more evolved compositions (matrix glass MgO concentrations down to 3.4 wt%, compared to 4.1 wt% in 2005 ('Methods')). Whole-rock and glass compositions erupted close to the caldera on F1 at the start of the eruption are relatively mafic, with average glass MgO concentrations of 5.0 wt% and low concentrations of incompatible elements (e.g., $P_2O_5 < 0.5$ wt% and $K_2O < 0.7$ wt%). In contrast, matrix glasses produced later in the eruption from the central segment of F1, and F4 and F5 (Fig. 1B) are more evolved (average MgO = 4.1 wt%), and with higher incompatible element concentrations.

Olivine–plagioclase–augite–melt (OPAM) barometry[33,34] performed on whole-rocks and glasses identify two distinct source depths ('Methods'). Material produced earlier in the eruption (F1 close to the caldera) equilibrated at a depth of $7.5 \pm 2.9$ km ($2.1 \pm 0.8$ kbar), whereas the later erupted, more evolved glasses (central F1, F4, and F5) equilibrated at $1.9 \pm 1.8$ km ($0.5 \pm 0.5$ kbar). The deeper equilibration depth is interpreted as indicating mid-lower crustal magma storage (Fig. 6E), consistent with the top of the low seismic velocity zone imaged through body wave tomography[35] and deeper earthquakes recorded between 2009 and 2011 (ref. [31]). The shallow equilibration depth is consistent with magma storage at the depth of pre-eruptive inflation estimated from the geodetic data, and the maximum depth of seismicity recorded from April 2018 through the eruption

**Fig. 6 Cartoon cross-sections depicting the inflation-deflation cycle of the 2018 eruption and plumbing system at Sierra Negra. A** Pre-eruptive inflation. **B** Mw5.4 earthquake. **C** Dyke intrusion. **D** Co-eruptive deflation, including schematic vertical (red arrows) and horizontal (black arrows) displacements of cGPS stations (green stars), earthquake hypocentres (blue circles), and possible geometry of magma plumbing system (orange for shallow equilibrated magmas, red for deep equilibrated magmas). Black dashed line provides a schematic illustration of relative change in surface shape during each interval. **E** Deeper magma plumbing system. Dashed lines indicate normalized kernel density estimates ("Methods") of equilibration depths for early (deep-red) and late (shallow-orange) erupted magmas.

(Figs. 5E and 6E). These storage depths are similar to other western Galápagos volcanoes[16,33].

## Discussion

The multiparametric observations of the 2018 eruption of Sierra Negra contrast markedly with those associated with the eruptions at Kilauea in 2018 and Bárðarbunga in 2014:

(1) The eruption at Sierra Negra was preceded by large amplitude, prolonged, and multi-staged uplift and intense accompanying seismicity.

(2) The eruption was triggered by intra-caldera faulting, in turn driven by elevated pressures in a shallow magma reservoir[36].

(3) The onset of the eruption involved complex magma ascent and lateral migration within the caldera. The main eruptive fissure system opened without detected seismicity, and only low rates of small earthquakes accompanied the NW flank intrusion.

(4) Co-eruptive subsidence was predominantly elastic, and of only slightly higher amplitude than the pre-eruptive uplift. Trapdoor faulting resulted in residual intra-caldera resurgence of 1.5 m.

These observations highlight fundamental differences between the processes driving eruption and caldera formation processes at Sierra Negra, and those at volcanoes with well-defined rift systems (e.g., Kilauea[37], Bárðarbunga[3], and Miyakejima[38]). We now discuss the implications in turn.

During the 13-year inflation process, supply of magma to the shallow sill drove predominantly elastic uplift of the caldera, and seismicity on the southern, western, and northern sections of the TDF (Fig. 6A). Pressurisation of the sill increased the stress on the TDF, eventually driving brittle failure and slip[36]. The 2018 eruption at Kilauea and 2014 eruption at Bárðarbunga were preceded by relatively minor uplift and seismicity, suggesting little additional magma was being supplied to the shallow plumbing system prior to eruption. At Sierra Negra, it has been hypothesised that slip on the TDF during inflation might act to increase the volume of the sill, reducing overpressure, and allowing the prolonged accumulation of magma[6,7,29]. However, we find no evidence from the cGPS displacement time series that earthquakes on the TDF influence either the uplift rate before the 2018 eruption or the subsidence rate during the eruption. Uplift and subsidence signals in the cGPS time series from stations in the centre of the caldera are linear before and after large TDF earthquakes.

The Mw5.4 earthquake at 09:15 on 26 June 2018 resulted in ~1.8 m of uplift along the southern TDF. In doing so, it unclamped the northern TDF, and promoted sill failure and magma ascent (Fig. 6B). At Kilauea and Bárðarbunga the main central reservoir drained laterally into established rift zones. At Kilauea, new activity initiated in the Middle East Rift Zone, associated with lateral down-rift dike propagation[1]. However, seismicity and deformation at Sierra Negra suggest bidirectional shallow magma migration, within or close to the northern and western sections of the TDF system (Fig. 6C). The eruption began on the northern rim of the caldera, near the eastward extent of seismicity on the northern TDF (Fig. 6D). Counterintuitively, OPAM barometry indicates that the first magma to erupt had a deep (~7.5 km) geochemical signal (Fig. 6E), so had only recently arrived at shallow levels. To explain these data, we suggest that the early stage of the eruption was supplied directly from the upper feeding system, rather than tapping the shallow-equilibrated magma stored in the sill itself. One possible geometry that would fit such observations is with the feeding system located towards the northern edge of the sill, close to the initial fissure systems at F1.

Evacuation of magma from the shallow sill drove caldera subsidence. Magma intruded aseismically into the northern flank, feeding eruptive fissure systems F1–3, resulting in the rapid deflation of the caldera in the first 24 h of the eruption. Magma also intruded into the NW flank, which later fed F4, triggered minor seismicity, and allowed continued subsidence of the caldera. All these subsequent eruptive fissures were fed with magma equilibrated to the 2 km depth of the shallow sill. However, despite renewed seismicity on the TDF, co-eruptive subsidence was predominantly elastic. This is in contrast to Kilauea and Bárðarbunga, where caldera collapse was accommodated by brittle faulting as magma flowed down rift.

Even though the inflation phase involved the largest pre-eruptive deformation and seismic signals reported from a basaltic volcano, the inflation–deflation cycle resulted in an absolute elastic subsidence in the centre of the caldera of 2 m. However, the slip difference between the opposite polarity 26 June Mw5.4 and 5 July Mw5.0 earthquakes resulted in net uplift of the sinuous ridge relative to the caldera rim of ~1.5 m, contributing to long-term resurgence. The formation of the 9.5 km by 7.5 km caldera at Sierra Negra is undated. Nevertheless, the initiation and subsequent movement on the TDF has resulted in growth of the

sinuous ridge above the elevation of the caldera rim (Fig. 1C). If 1.5 m is typical of the uplift over an eruption cycle, the average inter-eruption time is 14 years (based on the five confirmed eruptions since 1948), and the total uplift is now 150 m, the initiation of resurgence can be dated to ~1400 years ago. Caldera resurgence is a phenomenon typically found at silicic systems, therefore the occurrence of this style of deformation at a basaltic volcano sets it apart from other basaltic shield volcanoes.

Almost the entire TDF was seismically active during both uplift and subsidence. However, unlike Kilauea and Bárðarbunga, there is no evidence for activity on the caldera-bounding ring fault system. Instead, earthquake focal mechanisms and geodetically derived co-seismic displacements indicate translation and rotation of the trapdoor in addition to uplift (during inflation) and subsidence (during deflation). Similar rotational and translational kinematics have been observed on caldera-bounding faults at systems, including Miyakejima and Dolomieu calderas, and are suggested by numerical and analogue models[38]. The rapid final initiation of the eruption, and low levels of seismicity associated with migration of magma outside the caldera, means that forecasting and managing potentially hazardous future eruptions will be challenging, especially if magma were to intrude towards populated areas. Similar challenges should be expected at other basaltic shield volcanoes without well-developed rift zones

## Methods

**Seismic data and methods.** Seismic data were recorded by the permanent monitoring network of the IGEPN and two temporary deployments. Station GS09 of the SIGNET deployment (operational between late 2009 and 2011)[35] and station VCH1 (a Trillium compact 120; operational since 2012) of the IGEPN permanent network[39] were both installed at the same site close to the north rim of the caldera (Fig. 1).

In late April 2018, the IGUANA campaign network was installed at Sierra Negra (Supplemental Fig. 1). Installations were equipped with three different types of broadband seismometers, two units with Güralp 40 T, seven units with Güralp CMG-3ESPCD, and four units with Nanometrics Meridian Compact PH. The tiltmeter was of type Applied Geomechanics 701-2 A(4X). Data from the analogue tiltmeter and the 40 T seismometers were digitised and recorded with Güralp DM24S6EAM data-loggers. All seismic and tilt data were recorded with 100 samples per second. These data are also available from IRIS.

Data from stations GS09 and VCH1 are used to estimate the long-term seismicity rate at Sierra Negra (Figs. 2, 3, and 5). To detect local events, data are band-pass filtered between 5 and 10 Hz, and an STA/LTA algorithm is implemented in ObsPy[40]. A trigger is required on all three components of the single station data, and triggers with anomalous frequency content, very short trigger durations, or very short inter-event times are removed. We manually verify the high reliability of the algorithm by inspection of hundreds of events. For each event, we determine the root mean square amplitude of the vertical component of the velocity in a 10-s window around the peak amplitude, in a 1–20 Hz frequency band. We then estimate a local magnitude, based on calibration of the logarithm of the RMS amplitude and the magnitude of events co-reported in the IGUANA located catalogue from April 2018.

Seismicity rates and magnitudes estimated in this way assume that detections are associated with earthquakes located on the TDF system at Sierra Negra. We think this is a valid assumption, apart from known short-lived seismic episodes associated with eruptions at Fernandina in 2017 and 2018, and an intrusion at Cerro Azul volcano in 2017. Importantly, seismicity rate estimated in this way broadly follows that based on the less complete IGEPN catalogue of located events. The frequency content and s–p times of automatically detected events are similar to those recorded during the IGUANA deployment. Magnitudes estimated by this method assume events are equidistant from VCH1, a reasonable approximation for events located on the southern and western TDF, but likely overestimates the magnitudes of events on the northern TDF by ~0.5 units.

The high density of stations in the IGUANA network allowed high precision event locations. The earthquake catalogue was developed using SeisComP3, based on all near-caldera station data from April to August 2018, utilising the automatic detection and cluster-search location module scanloc in offline mode[41]. Only P wave arrivals were considered in the detection process. A manual scan of all station data for the period 26 June to 30 June was also undertaken. In the case of automatically detected events, P wave arrivals (picked on the vertical component) were manually reviewed and S wave arrivals (picked on the horizontal components) were manually picked where possible. This was done by a single operator, using both unfiltered and Butterworth filtered data. The use of unfiltered data was necessary due to the occurrence of clipped seismograms on some stations

for events with ML > 3.0. This yielded a total of 21,347 P arrivals and 3521 S arrivals, with an average pick uncertainty of 0.075 s.

The events were located using the location programme NonLinLoc[42], using the Oct-tree sampling algorithm and the one-dimensional velocity model[35]. Station elevation is taken into account by NonLinLoc, and a digital elevation model of the Galapagos Islands was used to further constrain earthquake hypocenters. The median horizontal error is 371 m and median vertical error is 672 m. Local Richter magnitudes, ML, for events in the IGUANA catalogue are calculated in SeisComP3, using the Hutton and Boore attenuation model for Southern California[43].

Focal mechanisms (Figs. 3–5) were determined from P wave first motions using the HaSH software[44] for 80 events recorded between the time of the installation of the IGUANA network to August 9, 2018. These earthquakes were selected to have approximately uniform spatial distributions along the TDF, and covering the pre- and co-eruptive phases. For each event, all available seismic records with a P wave polarity were picked (number of picks ranging from 7 to 19) and processed by the HaSH software, using a one-dimensional velocity model appropriate for the region.

**cGPS data and methods.** cGPS data used for this study came from a network of ten stations. The network was initiated in 2002 with a mix of two dual frequency and eight single frequency receivers. The network was then upgraded in 2010 to all dual-frequency receivers, collecting data every 30 s. The data were processed for daily static positions and high-rate (1 epoch/30 s) kinematic positions using the precise point positioning method implemented in the GIPSY-OASIS II version 6.3 software[45]. For the daily static positions, phase ambiguity resolution was performed using the single receiver algorithm[46], and we used final satellite ephemerides provided by the Jet Propulsion Laboratory (JPL). Final daily solutions were transformed into the IGb08 reference frame[47]. Kinematic analysis followed[48], where we applied ocean loading corrections using FES2004 (ref. [49]), modelled wet, and dry tropospheric zenith delays with VMF1 mapping functions[50]. All data are available through the UNAVCO archive: https://www.unavco.org/data/data.html.

The cGPS 30 s data were used to produce kinematic time series to investigate co-seismic displacements for pre- and syn-eruptive earthquakes and deformation within the Sierra Negra caldera related to magma migration. We estimated kinematic time series for all days with earthquakes >M4, and found that only earthquakes with $M > 4.5$ produced observable static co-seismic displacements. Co-seismic displacements were calculated by differencing the average of positions 1 h before and after the earthquake.

**InSAR data and methods.** The pre-eruptive deformation field in Fig. 2C is constructed from Sentinel-1 A, B data. Ninety-six data acquisitions were made from the end of 2014, using descending track 128. Interferograms were formed using the Sentinel-1 stack processor of the InSAR Scientific Computing Environment (ISCE) software[51,52], and the time series was obtained using the Miami INsar Time series software in Python[53].

The co-eruptive deformation field (Fig. 5C) is constructed using L-band (wavelength, ~22.3 cm) synthetic aperture radar (SAR) data from the Japan Aerospace Exploration Agency's ALOS-2 satellite. Data acquisitions were made on 18 May 2018 and 29 June 2018, using descending track T147, ScanSAR (WD1) acquisition mode, and incidence angle of 33 degrees. Interferograms were also formed using ISCE software. Topographic contribution to the interferometric phase was estimated, using a 12 m resolution DEM from the TanDEM-X mission. The interferogram was filtered using a power-spectral filter, filter strength 0.1, and unwrapped using the statistical-cost, network-flow algorithm for phase unwrapping, SNAPHU[52].

**Geodetic modelling of pre-eruptive inflation.** We compared our geodetic observations (cGPS time series—see above) of pre-eruptive inflation to the finite sphere[54] and sill-like[55] source geometries, using the dModels modelling software package[56]. Inversions were performed to estimate best-fit source location and depth ($x$, $y$, and $z$), along with the excess magma chamber pressure ($\Delta P/\mu$) and equivalent volume change. Poisson's ratio was set to 0.25 ($\lambda = \mu$), where $\lambda$ and $\mu$ are the first and second Lamé parameters. The location of the magma chamber at Sierra Negra has been constrained to the centre of the caldera and at 1.9–2.2 km depth inverting cGPS and InSAR data[6,7,22,25,57]. In order to compare current deformation signals with sources constrained in previous studies, an additional set of inversions was performed using sphere and sill geometries to solve for the same variables as before, but constrained at 2.1 km depth below the centre of the caldera ($-0.818°$ N, $-91.132°$ E).

**Stress modelling.** We used the Coulomb 3.4 software[58] to estimate the change in Coulomb failure stress and normal stress caused by the 09:15 UTC June 26, 2018 Mw5.4 earthquake that preceded the eruption by ~8 h. We used the cGPS derived co-seismic displacements for this earthquake and the earthquake focal mechanism to constrain the master fault geometry. We then estimated changes in Coulomb failure and normal stresses on faults with the orientation of the ring and TDF system on the north side of the caldera, and with the orientation of the initial eruptive fissure (fissure 1; Fig. 1). Stresses were calculated at depths of 0 and 2 km, the latter being the depth of the main pre-eruptive magmatic body determined geodetically and petrologically.

**Petrological data and methods**. Fresh lava and tephra samples were collected from close to the 2018 eruption vents during a series of fieldtrips in June–October 2018, permitted as part of Parque Nacional Galápagos project PC3018. The material was crushed in an agate mortar and pestle, and chips of matrix material were hand-picked, mounted in epoxy, ground, and polished for analysis. Microlite-free areas of matrix glass were identified by backscattered electron imaging, and analysed for major and trace elements using a Cameca SX100 electron microprobe (EPMA) in the Department of Earth Sciences, University of Cambridge. Glass analyses were performed using a 15 kV, 10 nA, defocused (12 μm) beam, with $SiO_2$ and alkalis analysed first to minimise the effect of electron-beam-induced sample damage. To ensure consistency between analytical sessions, glass data were internally calibrated using Smithsonian microbeam standard VG-2 (ref. [59]). Relative $2\sigma$ precision was estimated by repeat analysis of secondary standards and is better than ±5%, except for $K_2O$ (17.1%) and MnO (43.6%). $Cr_2O_3$ was typically below the detection limit.

Magma equilibration depths were calculated from glass and whole-rock data using the most recent recalibration of the OPAM barometer by Voigt et al.[34], which has been applied previously in Galapagos[33]. We use a melt $Fe^{2+}/(Fe^{2+} + Fe^{3+})$ ratio of 0.85 (approximating an oxygen fugacity at the quartz–magnetite–fayalite buffer)[60] and assume negligible Cr, based on our EPMA data. As the low crystallinity of Sierra Negra samples precludes visual determination of olivine, plagioclase, and augite co-saturation, we assess three-phase saturation using a statistical approach[61], accepting barometric results with probability factors ($P_F$) > 0.8. This method likely rejects some false positives, but minimises the model uncertainty[61]. A standard estimate of error has not yet been determined for the Voigt et al.[34] OPAM calibration but we conservatively assume it is equal to earlier models (±1.4 kbar)[62]. We evaluate our barometric results using kernel density estimates, with bandwidths calculated after Sheather and Jones[63]. As the crustal velocity profile and Moho depth are similar in western Galapagos and the Big Island of Hawaii[64,65], we convert magma equilibration pressures to depth using the polynomial relationship of Putirka[66].

Major and trace element compositions (Supplemental Fig. 4) were determined at the PSO/IUEM (Pôle Spectrométrie Océan, Institut Universitaire Européen de la Mer, Brest, France), following the analytical procedure of Cotton et al.[67]. Typically 250 mg of rock powder were dissolved in closed screw-top teflon vessels (Savillex) at about 90 °C for 1 day using 3 ml of concentrated HF, and 1 ml of concentrated $HNO_3$. Next, 96 ml of $H_3BO_3$ aqueous solution (20 g/l $H_3BO_3$) were added to neutralise the excess HF. All reagents used are analytical grade.

Elements were measured by inductively coupled plasma–atomic emission spectrometry, using a Horiba Jobin Yvon® Ultima 2 spectrometer. The boron included in the solution was used as an internal standard. Calibrations were made using international standards, ACE, ME, WSE, and JB2. For major elements, relative standard deviation is ≤1% for SiO2 and ≤2% for the other major elements, for trace elements standard deviation is ≤5%.

## Data availability

Earthquake catalogues, cGPS time series, and interferograms underpinning these results are available on Zenodo at https://doi.org/10.5281/zenodo.4389190. Raw seismic data from IGUANA project are available from the Incorporated Research Institutions for Seismology (IRIS), Data services (https://doi.org/10.7914/SN/8G_2018) and data from the Ecuadorian national network (https://www.fdsn.org/networks/detail/EC/) on reasonable request from the IGEPN at https://www.igepn.edu.ec/datos-mseed. Raw cGPS data are available through the UNAVCO archive: https://www.unavco.org/data/data.html, at: https://doi.org/10.7283/ESJY-V915, https://doi.org/10.7283/FE83-C961, https://doi.org/10.7283/T5N58JM5, https://doi.org/10.7283/FY2B-DV40, https://doi.org/10.7283/HB35-V642, https://doi.org/10.7283/QBZB-Z543, https://doi.org/10.7283/T5M61HHS, https://doi.org/10.7283/61RQ-0887, https://doi.org/10.7283/WSXX-VM46, https://doi.org/10.7283/T5CN725H, https://doi.org/10.7283/GY33-F024, https://doi.org/10.7283/XZAA-W505, https://doi.org/10.7283/T5445JQC, and https://doi.org/10.7283/S8FW-YS19. Sentinel-1 raw SAR data that support the findings of this study are publicly available at https://scihub.copernicus.eu. ALOS-2 raw SAR data availability is restricted to PI investigation at www.eorc.jaxa.jp/ALOS/en/.

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

## Acknowledgements

The authors thank the IGEPN, the Galápagos National Park, Charles Darwin Research Station, staff at IGEPN for sample collection, Iris Buisman for technical assistance with EPMA, and I.G. Main for helpful comments on the manuscript. This material is based on services provided by the GAGE Facility, operated by UNAVCO, Inc., with support from the National Science Foundation and the National Aeronautics and Space Administration under NSF Cooperative Agreement EAR-1724794. Part of this work was funded by the NSF award EAR 1838373, the IGUANA NERC Urgency grant NE/S002685/1, the Marshall-Heape and Vokes fellowship funds, Tulane University, a Laboratoire Mixte International Séismes et Volcans dans les Andes du Nord supported by the Institut de Recherche pour le Développement, and an EPN grant PIJ-18-02 for B.B. Part of this research was carried out at the Jet Propulsion Laboratory, California Institute of Technology, under a contract with the National Aeronautics and Space Administration (grant 281945.02.47.04.51).

## Author contributions

A.F.B., P.C.L., and M.R. led and coordinated the research, field activities, and writing. A.F.B., C.J.B., C.E., J.G., S.H., M.M., S.-J.O., and M.R. contributed to seismic data collection, processing, analysis, and interpretation. P.C.L., M.H., N.M., and A.G.R. contributed to cGPS data collection, processing, analysis, and interpretation. F.A., M.B., and P.L. contributed to InSAR data processing, analysis, and interpretation. B.B., C.L., M.G., and M.J.S. contributed to geochemical data processing, analysis, and interpretation. All authors contributed to the writing of the manuscript.

## Competing interests

The authors declare no competing interests.
