## [Peer Review File · Nature Communications]

REVIEWER COMMENTS

Reviewer #1 (Remarks to the Author):

This is a well-crafted paper that conveys a very important dataset from the 2018 eruption of Sierra Negra, Galapagos. What is unique about the paper is the extent to which multiparametric data (deformation, seismicity, geochemistry) are integrated to achieve a detailed, step-by-step documentation of the 2018 eruption and the response of the caldera to the eruption, as well as the role of the trap-door fault in the process. This paper has the potential to be a benchmark publication in multidisciplinary studies of eruptive processes, given the quality and extent of the dataset.

In the interest of full disclosure, I am a geochemist, not a geophysicist or a geodynamicist, so I cannot comment on the methodology for the bulk of the work here, nor the accompanying modeling efforts. As someone who works in Galapagos on eruptive activity, all of the findings are reasonable and the data support them insofar as I can tell. The geochemical aspects of the study are not used to their fullest advantage (my bias, of course), nor explored in as much depth as they could be (given that comparable data exist for the 2005 eruption at Sierra Negra, and some for the 1979 eruption). What do we learn about the system from changes that occurred in equilibrium depth from 1979 to 2005 to 2018, for instance, or from extent of fractionation? It'd be really interesting to know more about the nature of the magma supply and its source, but that is beyond the scope of this study for sure.

I have made a number of absolutely minor suggestions below (numbered in no particular order) for improving clarity and catching minor typos.

The only substantive suggestion I have of this manuscript is that the authors are not really highlighting the importance and rarity of what they have done here, nor are they maximizing the discussion about its potential impact on a more global scale. The multiparametric nature of this study is truly rare, and even more so at basaltic systems; a short additional paragraph or two that considers(s) these findings about caldera dynamics relative to other ocean island systems, for instance, would broaden the potential impact of this manuscript. Alternatively, a short discussion about how the combination of these multiparametric data reveals more than we can learn from any of the sub-disciplines/datasets alone would be equally useful and increase the utility of the paper beyond being more regional in scope. Similarly, more contextualization about the relative rarity (or commonality?) of trap-door faults and their role in eruptive processes globally would be appropriate. Adding at least one of these discussion components would give the manuscript considerably more impact, would draw more readers, and would highlight the excellent and

important research done herein to a significantly greater extent, in addition to giving the manuscript more of the “punch” that one generally expects from Nature Geoscience papers.

Minor comments:

1. The abstract could use a bit more polishing. It takes rereading a few times to put the 1st reference to 1.5 m uplift into context, for instance.
2. It strikes me as odd that the Charles Darwin Research Station and staff are not mentioned in the acknowledgment; they are a separate entity from the Galapagos National Park, and do much of the support work when researchers work in the archipelago. This must be an oversight?
3. P.3 line 55: should state whether 150 m TDF is height/offset or length of sinuous ridge to be clear.
4. Line 92: Verb disagreement with subject (distribution), just FYI. Also on line 289 (assume).
5. Line 400: standard estimate of error, not error of estimate.
6. Line 404: Where in Hawaii? Presumably the Big Island?
7. Line 409: Missing the authors' names Cotton et al..
8. Line 410: ml should be mL
9. Figure 1: Should have closer view of caldera to show extent of sinuous ridge. Is the dashed line in 1B really the TDF? It maps more logically as the sinuous ridge.
10. Figure 2: What is red line in B (average number of seismic events presumably, but should be indicated). Also the green and red dashed lines for the 2005 eruption and the later intrusion are not easy to read or notice, and should be called out in the caption; intrusion year label should be moved, can't see the critical 1 of 2010.
11. The phases in Figure 2 are not referenced by name in the manuscript, which would help make the detailed descriptions easier to follow and digest for the reader. Otherwise, why name them in the figure?
12. Figure 3: This is all very condensed and hard to extract, particularly the “colored by time” data in B and C. The red line needs to be defined in B. It's really tough to extract any pattern from the events colored by time in C.
13. Figure 4: Were the lava flows erupted only after 12PM? That might be worth clarifying (that they are represented accurately in the time slices shown in D and E parts of the figure). I might suggest (if feasible) that some kind of animation to show the shifts in locations of earthquakes and deformation over time could be really effective (more so than Figure 4, which is tough to interpret).
14. Figure 5: Again, some kind of animation over time would be so helpful. Even just layering the different time slices on the figure one after the other and grabbing it as a Quicktime screen recording would be a nice addition to help the reader interpret these (very busy) figures.

15. Figure 6: In E, I think deep and shallow in the text need to be orange and red, not blue and black. Need to convey direction of the cross sectional line (SW to NE again, presumably). Also, what are the dashed black lines over top of each surface in A-D? Change in surface shape? Sorry if I missed this.

16. Discussion opening paragraph: I know such multiparametric observations have been collected on such active volcanoes as Kilauea, but it would be helpful for the authors to convey a few other locations, so that the reader can put into context the significance of this being a first for the Galapagos (what I mean is, how rare is it to get this level of data collection for an eruption?). Also, the second sentence in this pgh needs polishing, it doesn't read smoothly.

17. Line 208 verb disagreement (is not/are not)

18. The geochemical data are not being used to a significant extent, beyond determining equilibration depths for two major phases of the erupted material. Additional consideration of why the extent of differentiation varied as it did would be helpful and appropriate here. For instance, why are the lavas more evolved than what was erupted in 2005? What processes might be responsible for this shift?

19. Line 232: suggestion that feeding system of sill is on the north side of the caldera has little to no basis (as written). What is the evidence that moving the feeding system to the south side of the caldera would have resulted in a different geochemical signature? This is rather speculative, and needs more support/justification.

20. Pgh beginning line 236: Is there no deformation data that can be brought into this analysis from either this eruption of Bardarbunga's?

21. This would be beyond my artistic or special visualization abilities, but some kind of more 3-dimensional depiction of the pre- and syn-eruption events and the role of the trapdoor fault in them would be really enlightening here, in place of the final figure (or as a supplement to it).

This is critically important work that integrates observations from multiple disciplinary perspectives, a collaborative approach to research that we should all strive to achieve when studying complex natural systems.

--Karen Harpp, Colgate University

Reviewer #2 (Remarks to the Author):

There have been several major eruptions and caldera collapse events at basaltic shield volcanoes in the last few years, including most notably at Kilauea, Hawaii and Bardarbunga, Iceland. This manuscript describes the most recent in this series, the 2018 eruption and caldera collapse at Sierra Negra, Galapagos. The manuscript describes the eruption using a combination of seismic, geodetic and petrological observations. The methods are state-of-the-art and clearly described, but not novel. The analysis is thorough and the manuscript culminates in a detailed description of the sequence of events.

However, there is no attempt at modelling the system as a whole (for example, to estimate reservoir failure thresholds as done for Kilauea by Anderson et al, 2019 and Bardarbunga by Gudmundsson et al, 2016). The result is a very detailed description of an interesting event, which I expect this could form the basis for some interesting insights in future studies, but in its current form doesn't provide any novel insights into the processes responsible.

To summarise, this manuscript presents a combination of different datasets and analysis applied to an interesting recent eruption and is therefore timely. The writing is clear and the analysis is thorough. There is nothing particularly controversial. As such, I don't have any minor comments. However, to appeal to the broad readership of Nature Communications, I think some greater scientific insights would be needed. Some possible avenues to explore would be a comparison with the other major caldera collapse events that have occurred recently, or a quantitative assessment of the conditions required for collapse. To me, an interesting difference between this and the other recent caldera collapse events is the rapid and prolonged uplift that occurred prior to the eruption

Major revisions in response to reviewers' comments

Broader implications of results for basaltic volcanoes:

Both reviewers make the same major comment; that they feel we could make more of our detailed observations in terms of a more general understanding of basaltic systems. On re-reading the manuscript, we feel that this point is fair and, in response, we have completely rewritten the discussion section of the paper to this end. The discussion now describes how the pre-eruptive inflation, complex eruption initiation process, and co-eruptive deflation, differs markedly from that seen at Kilauea in 2018 and at Bardarbunga in 2014. We argue that the absence of rift zones at Sierra Negra provides a first order control on these differences.

In the new discussion section we first summarise the differences in lines 207-221, and then discuss them further in sections on 'Pre-eruptive inflation' (lines 223-236), 'Eruption onset' (lines 239-252), and 'Co-eruptive deflation' (lines 254-262).

We do politely contest, however, Reviewer #2's assertion that the study 'doesn't provide any novel insights into the processes responsible'! The manuscript is based largely on observational data rather than model results, but these are rich in novel insights. Our manuscript presents and describes new insights into many eruptive processes seen here for the first time through integration of seismic and geodetic data. As well as being the first *local* multiparametric observations of an eruption at a Galapagos Island basaltic volcano, they are the first of an eruption anywhere involving trapdoor faulting (the 2014 eruption and caldera formation at Bardarbunga involved 'trapdoor faulting' but was not well-observed by local networks due to ice cover), and the first that results in caldera resurgence. Just by way of example, novel insights include:

- (a) the entire trapdoor fault, minus the northeast hinge zone, is seismically active during pre-eruptive inflation and co-eruptive deflation, yet the caldera the caldera bounding fault is not active,
- (b) during eruption onset, deformation suggests magma migrates in two directions, triggering seismicity around the trapdoor fault, and that magma movement is almost aseismic outside the caldera,
- (c) an increment of resurgence is controlled by the net slip between the largest inflationary and deflationary trapdoor faulting earthquakes, even though total deflation results in elastic subsidence of the caldera, not brittle failure on the caldera bounding faults.

We indeed hope that future studies will use our datasets as the basis for quantitative modelling of parts or all of the system, but do not feel that diminishes the significance of the insights resulting from these primary observations.

Geochemical aspects of the study:

Prof Harpp suggests we include a comparison of geochemical data from the 1979, 2005, and 2018 eruptions. We fully agree that this would be an interesting comparison, but to do this justice would be a manuscript in itself, and so we feel that it is beyond the scope of this current paper. Our focus here is on caldera dynamics, and the insights that we can derive from multiparametric datasets. Its strength comes from combining these different datasets, but limits the detail we can go in to for

each individual datasets. For example, there are aspects of the seismicity that we consider to be beyond the scope of this specific study, and that we intend to address elsewhere.

Response to Prof. Harpp's minor comments

1. The abstract could use a bit more polishing. It takes rereading a few times to put the 1st reference to 1.5 m uplift into context, for instance.

We have substantially rewritten the abstract in the revised manuscript and hope it is now easier to read.

2. It strikes me as odd that the Charles Darwin Research Station and staff are not mentioned in the acknowledgment; they are a separate entity from the Galapagos National Park, and do much of the support work when researchers work in the archipelago. This must be an oversight?

An acknowledgment to the Charles Darwin Research Station has been added.

3. P.3 line 55: should state whether 150 m TDF is height/offset or length of sinuous ridge to be clear.

Line now reads 'The TDF has produced a 150 m high 'sinuous' ridge...'

4. Line 92: Verb disagreement with subject (distribution), just FYI. Also on line 289 (assume).

Both sentences now corrected

5. Line 400: standard estimate of error, not error of estimate

Sentence now corrected

6. Line 404: Where in Hawaii? Presumably the Big Island?

Yes, 'Big Island' now added to sentence.

7. Line 409: Missing the authors' names Cotton et al.

Names added

8. Line 410: ml should be mL

Corrected

9. Figure 1: Should have closer view of caldera to show extent of sinuous ridge. Is the dashed line in 1B really the TDF? It maps more logically as the sinuous ridge.

The details of the intra-caldera fault system are already well described in the literature (e.g. Jonsson et al., 2005), and are not really the main focus of this paper. At the scale of Fig. 1B, the dashed line is

effectively representing both the TDF and Sinuous Ridge – although future discussions about how the new seismicity data sheds light on the nature of, and relationship between, these two features.

10. Figure 2: What is red line in B (average number of seismic events presumably, but should be indicated). Also the green and red dashed lines for the 2005 eruption and the later intrusion are not easy to read or notice, and should be called out in the caption; intrusion year label should be moved, can't see the critical 1 of 2010.

Legend modified to explain these features of the figure. Text moved from Fig. 2 A to Fig. 2B for clarity.

11. The phases in Figure 2 are not referenced by name in the manuscript, which would help make the detailed descriptions easier to follow and digest for the reader. Otherwise, why name them in the figure?

We have modified the text to now reference the uplift phases. The text now reads 'with at least four distinct uplift episodes (Phases 1-4 in Fig. 2) and one subsidence episode (Methods). From late 2017 until the onset of the eruption, during the latter stages of the fourth uplift phase,...'

12. Figure 3: This is all very condensed and hard to extract, particularly the "colored by time" data in B and C. The red line needs to be defined in B. It's really tough to extract any pattern from the events colored by time in C.

The red line is now defined in the caption of Figure 3. Part of the point of Fig. 3C is that there is no pattern in the location of the events with time. To make this clear, we have added an explanatory sentence to the text 'There is no systematic evolution of the location of these earthquakes along the TDF with time.' To help demonstrate this, we have included an *animation* of these data in the supplemental material.

13. Figure 4: Were the lava flows erupted only after 12PM? That might be worth clarifying (that they are represented accurately in the time slices shown in D and E parts of the figure). I might suggest (if feasible) that some kind of animation to show the shifts in locations of earthquakes and deformation over time could be really effective (more so than Figure 4, which is tough to interpret).

Yes, as far as we can ascertain, the first lava flows erupted at ~19:35. We have included two animations of the data as supplemental material – one of the April-August 2018 period, and one of the day of eruption onset. We hope these are helpful for interpreting these complex data.

14. Figure 5: Again, some kind of animation over time would be so helpful. Even just layering the different time slices on the figure one after the other and grabbing it as a Quicktime screen recording would be a nice addition to help the reader interpret these (very busy) figures.

Please see response to point 13 above

15. Figure 6: In E, I think deep and shallow in the text need to be orange and red, not blue and black. Need to convey direction of the cross sectional line (SW to NE again, presumably). Also, what are the dashed black lines over top of each surface in A-D? Change in surface shape? Sorry if I missed this.

Corrected in the legend. Yes, dashed black line is a schematic indication of the change in surface shape.

16. Discussion opening paragraph: I know such multiparametric observations have been collected on such active volcanoes as Kilauea, but it would be helpful for the authors to convey a few other locations, so that the reader can put into context the significance of this being a first for the Galapagos (what I mean is, how rare is it to get this level of data collection for an eruption?). Also, the second sentence in this pgh needs polishing, it doesn't read smoothly.

The Discussion has now been completely rewritten with this in mind. Please see response to major comments above, and new discussion.

17. Line 208 verb disagreement (is not/are not)

This text is not present in the revised manuscript

18. The geochemical data are not being used to a significant extent, beyond determining equilibration depths for two major phases of the erupted material. Additional consideration of why the extent of differentiation varied as it did would be helpful and appropriate here. For instance, why are the lavas more evolved than what was erupted in 2005? What processes might be responsible for this shift?

Yes, this is true. See comment above regarding the breadth of the article.

19. Line 232: suggestion that feeding system of sill is on the north side of the caldera has little to no basis (as written). What is the evidence that moving the feeding system to the south side of the caldera would have resulted in a different geochemical signature? This is rather speculative, and needs more support/justification.

This is indeed speculative. However, the plumbing system needs to be located somewhere, and there is no direct evidence to constrain it. There is plenty of evidence from other systems (e.g. the Isle of Rum) that the feeding system can be off-centred from a broadly circular sill. We know that the first magmas to erupt, on the caldera rim, had recently arrived from the deep reservoir. However, there is no indication in the seismic or geodetic data (and probably not enough time) for them to have risen between the TDF earthquake and the start of the eruption. Instead, it seems likely that the early phase of the eruption was tapping magma that had recently arrived at the shallow reservoir. Perhaps the simplest way this could happen is if the feeding system was located in the north of the caldera, near to where the eruption started. This argument does make some assumptions about the nature of magma mixing within the sill, but in the absence of other evidence, seems the easiest way to explain the observations.

We have expanded the text in new lines 242-257 in the Discussion to now say:

'Counterintuitively, the first magma to erupt had a deep (~7.5 km) geochemical signal (Fig. 6E), so had only recently arrived at shallow levels. To explain these data, we suggest that the early stage of the eruption was supplied directly from the upper feeding system, rather than tapping the shallow-equilibrated magma stored in the sill itself. One possible geometry that would fit such observations is with the feeding system located towards the northern edge of the sill, close to the initial fissure systems at F1.'

20. Pgh beginning line 236: Is there no deformation data that can be brought into this analysis from either this eruption of Bardarbunga's?

This paragraph is no longer present in the new discussion, though I'm not quite sure what you mean by the question. Is this in terms of identifying/forecasting aseismic magma movement (in which case, yes, if the cGPS or tilt stations happen to be located in the right place).

21. This would be beyond my artistic or special visualization abilities, but some kind of more 3-dimensional depiction of the pre- and syn-eruption events and the role of the trapdoor fault in them would be really enlightening here, in place of the final figure (or as a supplement to it).

It's beyond ours too! We hope that the new animations help in this visualization. We have spent a long time thinking about how to best present these complex 3D data, and it really isn't easy.

REVIEWER COMMENTS

Reviewer #1 (Remarks to the Author):

Review of revised version of Bell et al. Nature Geoscience caldera resurgence Sierra Negra
November 2020

I appreciate the detailed response to my initial review comments and concur with the authors on all of them but one (very minor): it would be helpful to add half a sentence that justifies how you know that the first magma has a deep geochemical signal (what's the geochemical evidence)? That would help support the contention that it's deep, and in turn would support your proposed mechanism a bit more too.

The abstract is definitely improved and more impactful than its original version.

Line 49: distinguish them, rather than distinguishes them

Line 68: Low roof thickness?

Line 276: Phenomenon, not phenomena

Line 284: Sentence beginning with "Similar kinematics" needs to be more specific—what other caldera systems (some examples)? Otherwise this sentence just isn't adding much clarity to the discussion.

Last line: I would remove "southward" or add a second sentence. The "southward" suggests you are only talking about Sierra Negra (I might specify also Villamil, since it's not exactly a lot of populated areas, but just one relatively small town and surrounding agricultural regions); it would be more impactful to end the main body of the paper with a broader statement, about how this and presumably other volcanoes that are without rift zones pose significant predictive challenges given the seismic migration of magma outside the caldera. In case I didn't explain that well, I would just recommend broadening the last sentence to reach beyond just Sierra Negra, which would also tie it in more coherently to the introduction.

The videos are indeed very helpful, and I fully realize how complex it is to figure out how to depict all this effectively! Thank you for adding it. I often think about papers from the perspective of early grad and senior undergraduates reading them, b/c I rely on Nature Geoscience among a few other journals for the sources of these types of learning tools. With something that makes visualization of the data easier for a relative beginner, it makes the paper more useful as a pedagogical resource. One minor suggestion: Add the legend for the significance of the various dot colors on the video in addition to the scale for deformation. People will inevitably click on this early in reading the manuscript, and allowing them to grasp its significance immediately would keep them more engaged in the paper, I suspect. These are slightly tough videos to interpret with the simultaneous horizontal and vertical arrows; I realize this would be challenging, but some kind of 3-dimensional depiction of the deformation would still be ideal (one that shows the surface deforming in both the horizontal and vertical dimensions). That said, these additional videos are a significant improvement over the static figures from the first submission, and the combination of the seismic data with the deformation is quite impactful.

Bottom line: This is an important, potentially highly impactful paper for our understanding of the mechanisms of caldera resurgence and eruption at basaltic volcanoes, and I strongly recommend its publication, pending these (super minor) suggestions I've made above. Very exciting work!

Sincerely,

Karen Harpp, Colgate University

Reviewer #2 (Remarks to the Author):

This manuscript provides a detailed description of events leading up to and during the 2018 eruption of Sierra Negra in the Galapagos. As my initial review stated, the methods and observations are well-described and state-of-the-art and this remains the case in the revised manuscript.

The revised manuscript goes significantly beyond the initial submission by putting the observations in context of other large basaltic eruptions that have occurred in recent years (most notably at Bardarbunga and Kilauea). The result is that the novelty of the observations is much clearer, as are the new insights into the processes occurring at basaltic calderas. I would like to thank the authors for their polite rebuttal and congratulate them on a much improved paper, which I am now convinced would be of interest to a broad authorship.

My comments on this version of the manuscript focus on just 2 areas:

The first point is related to the intrusion to the NW, which is an interesting and a rather neglected part of the eruption story. Although Sierra Negra doesn't have well-developed rift zones like Bardarbunga and Kilauea, this is evidence that magma withdrawal feeding lateral intrusion remains a key part of the system. All that is presented is a single ALOS2 interferogram. However, Sentinel-1 data is available for this time period and would give a time-series with 6 day resolution. Indeed a quick look at our own automatically generated data suggests that the uplift of ~ 1 m was over by 1st July, followed exponentially decaying subsidence which extended beyond the end of the eruption in August and totalled 2m by 2020. This contrasts with the GPS inside the caldera which shows subsidence during the intrusion which had tailed off by the end of August (as far as I can see from Fig 5A). This raises some interesting questions regarding magma budgets, and the process responsible for subsidence within the caldera.

Even if a more detailed analysis is not included, the interferogram shown in Fig 5D needs to be described more fully. What is the maximum amount of deformation? Does the geometry of the intrusion correspond to a sill or a dyke (or something twisted as previously observed in the Galapagos)? The deformation pattern is continuous, so can this really be characterized as two distinct intrusions. Although there is less seismicity in this region, it is not true to say that it is aseismic (or even 'largely' aseismic) without some quantitative estimates.

Secondly, I agree that it is very challenging (and time-consuming) to represent the richness of these multiparameter datasets using just a small number of 2D plots. The animations are a useful addition, but need better explanation (and preferably stand-alone). For example, what are the green stars? I assume the colour of the dots is the time of the earthquake and the size represents the magnitude, but no key is provided. One movie starts with no GPS displacements (in June), but the other starts in April and already shows several meters of uplift. I assume the difference is the reference, but that isn't explained. Neither movie incorporates information about the eruption itself (timing, location etc). Even if the focus is the seismic and geodetic observations, some context is needed for the general reader not familiar with the timeline.

There are also a couple of minor points on the figures:

Figure 4 – The location of GV03 and GV09 are shown in panel E, but not SN14.

Figure 5 – does the size of the symbols represent earthquake magnitude? No key provided.

Response to Prof. Harpp's comments

1. *'...it would be helpful to add half a sentence that justifies how you know that the first magma has a deep geochemical signal (what's the geochemical evidence)? That would help support the contention that it's deep, and in turn would support your proposed mechanism a bit more too.*

We point the reviewer to the paragraph starting at line 197, which describes the OPAM geobarometry results. We have edited the text on lines 249-251 to now read: 'Counterintuitively, OPAM barometry indicates that the first magma to erupt had a deep (~7.5 km) geochemical signal (Fig. 6E), so had only recently arrived at shallow levels.'

2. *Line 49: distinguish them, rather than distinguishes them*

Corrected

3. *Line 68: Low roof thickness?*

Corrected

4. *Line 276: Phenomenon, not phenomena*

Corrected

5. *Line 284: Sentence beginning with "Similar kinematics" needs to be more specific—what other caldera systems (some examples)? Otherwise this sentence just isn't adding much clarity to the discussion.*

Sentence edited to now say: 'Similar rotational and translational kinematics have been observed at other caldera systems including Miyakejima and Dolomieu calderas, and are suggested by numerical and analogue models³⁸'

6. *Last line: I would remove "southward" or add a second sentence. The "southward" suggests you are only talking about Sierra Negra (I might specify also Villamil, since it's not exactly a lot of populated areas, but just one relatively small town and surrounding agricultural regions); it would be more impactful to end the main body of the paper with a broader statement, about how this and presumably other volcanoes that are without rift zones pose significant predictive challenges given the seismic migration of magma outside the caldera. In case I didn't explain that well, I would just recommend broadening the last sentence to reach beyond just Sierra Negra, which would also tie it in more coherently to the introduction.*

We removed 'southwards' from the sentence and added a further sentence: 'Similar challenges should be expected at other basaltic shield volcanoes without well-developed rift zones'.

7. The videos are indeed very helpful, and I fully realize how complex it is to figure out how to depict all this effectively! Thank you for adding it. I often think about papers from the perspective of early grad and senior undergraduates reading them, b/c I rely on Nature Geoscience among a few other journals for the sources of these types of learning tools. With something that makes visualization of the data easier for a relative beginner, it makes the paper more useful as a pedagogical resource. One minor suggestion: Add the legend for the significance of the various dot colors on the video in addition to the scale for deformation. People will inevitably click on this early in reading the manuscript, and allowing them to grasp its significance immediately would keep them more engaged in the paper, I suspect. These are slightly tough videos to interpret with the simultaneous horizontal and vertical arrows; I realize this would be challenging, but some kind of 3-dimensional depiction of the deformation would still be ideal (one that shows the surface deforming in both the horizontal and vertical dimensions). That said, these additional videos are a significant improvement over the static figures from the first submission, and the combination of the seismic data with the deformation is quite impactful.

Animating a 3D surface through the deformation data is sadly beyond my expertise at the moment. As a compromise, we have added time series graphs to the animations to broaden their scope, and act as legends to the map component. We have also improved the caption to go along with these animations.

Response to Reviewer #2's comments

1. *The revised manuscript goes significantly beyond the initial submission by putting the observations in context of other large basaltic eruptions that have occurred in recent years (most notably at Bardarbunga and Kilauea). The result is that the novelty of the observations is much clearer, as are the new insights into the processes occurring at basaltic calderas. I would like to thank the authors for their polite rebuttal and congratulate them on a much improved paper, which I am now convinced would be of interest to a broad authorship.*

We thank the reviewer for their thoughtful comments.

2. *The first point is related to the intrusion to the NW, which is an interesting and a rather neglected part of the eruption story. Although Sierra Negra doesn't have well-developed rift zones like Bardarbunga and Kilauea, this is evidence that magma withdrawal feeding lateral intrusion remains a key part of the system. All that is presented is a single ALOS2 interferogram. However, Sentinel-1 data is available for this time period and would give a time-series with 6 day resolution. Indeed a quick look at our own automatically generated data suggests that the uplift of ~ 1m was over by 1st July, followed exponentially decaying subsidence which extended beyond the end of the eruption in August and totalled 2m by 2020. This contrasts with the GPS inside the caldera which shows subsidence during the intrusion which had tailed off by the end of August (as far as I can see from Fig 5A). This raises some interesting questions regarding magma budgets, and the process responsible for subsidence within the caldera.*

Even if a more detailed analysis is not included, the interferogram shown in Fig 5D needs to be described more fully. What is the maximum amount of deformation? Does the geometry of the intrusion correspond to a sill or a dyke (or something twisted as previously observed in the Galapagos)? The deformation pattern is continuous, so can this really be characterized as two distinct intrusions. Although there is less seismicity in this region, it is not true to say that it is aseismic (or even 'largely' aseismic) without some quantitative estimates.

We thank the reviewer for their comment regarding the northwest intrusion. We agree that a detailed analysis is outside the scope of this article, and refer the reviewer to an article by co-authors describing the dynamics of this intrusion here:

<https://eartharxiv.org/repository/view/1784/>.

Maximum deformation along the intrusion to the NW as measured by ALOS-2 is ~2.0 m in the line-of-sight direction (incidence angle 33 degrees) and mostly occurred during the propagation phase between June 26th and July 1st. Similarities between ascending and descending data (not presented in the manuscript) allow us to infer that this deformation is predominantly vertical. The maximum displacement cannot be inferred from the Sentinel-1 data due to the extreme deformation. We looked at the evolution of the deformation along the intrusion using Sentinel-1 6-day-repeat data, as described by the reviewer, and indeed you could see an exponentially decaying subsidence. Some of this subsidence could have been easily related to the intensity of the eruption waning but some of it, especially after the eruption ended, could be related to the contraction of the intrusion as it cools/outgases. Correlating caldera subsidence with the opening and closing of the intrusion is outside the scope of this article.

Response: We have added the following language to lines 268-269 to describe the deformation associated with the NW sill: “Analysis of ALOS-2 data indicate ~2 m of uplift associated with the intrusion during the propagation phase.”

The main eruptive fissures opened with pretty much no detected seismicity, so I think it is reasonable to refer to the northerly intrusion as ‘largely aseismic’ – certainly relative to the processes happening in the caldera. The seismicity associated with the more westerly intrusion is very modest, though I accept that it is a stretch to refer to this as ‘aseismic’. Lines 217-218 have now been rewritten to read: ‘The main eruptive fissure system opened without detected seismicity, and only low rates of small earthquakes accompanied the NW flank intrusion’. Lines 289-291 have been rewritten to say ‘The rapid final initiation of the eruption, and low levels of seismicity associated with migration of magma outside the caldera ...’

3. Secondly, I agree that it is very challenging (and time-consuming) to represent the richness of these multiparameter datasets using just a small number of 2D plots. The animations are a useful addition, but need better explanation (and preferably stand-alone). For example, what are the green stars? I assume the colour of the dots is the time of the earthquake and the size represents the magnitude, but no key is provided. One movie starts with no GPS displacements (in June), but the other starts in April and already shows several meters of uplift. I assume the difference is the reference, but that isn’t explained. Neither movie incorporates information about the eruption itself (timing, location etc). Even if the focus is the seismic and geodetic observations, some context is needed for the general reader not familiar with the timeline.

Response: We have added time series of earthquake magnitudes and deformation in both animations to provide additional information and to act as legends for the symbols on the map. We have also provided captions for the animations in the Supplemental information.

4. There are also a couple of minor points on the figures:

Figure 4 – The location of GV03 and GV09 are shown in panel E, but not SN14.

There isn’t space to easily label tiltmeter SN14 on Fig. 4E – it is very close to GV09. The caption now refers the reader to Fig 1E, where SN14 is labelled.

Figure 5 – does the size of the symbols represent earthquake magnitude? No key provided.

Yes, earthquake symbol size and colour represent magnitude and time throughout, and are as indicated in Fig. 5B. This has now been added to the caption for Fig. 5

REVIEWERS' COMMENTS

Reviewer #2 (Remarks to the Author):

Thank you for addressing those minor revisions. I have no further comments.